# Modelling the contribution of the hypnozoite reservoir to *Plasmodium vivax* transmission

**Michael T White[1]\*, Stephan Karl[2,3], Katherine E Battle[4], Simon I Hay[4,5], Ivo Mueller[2,3,6], Azra C Ghani[1]**

[1]MRC Centre for Outbreak Analysis and Modelling, Department of Infectious Disease Epidemiology, Imperial College London, London, United Kingdom; [2]Department of Infection and Immunity, Walter and Eliza Hall Institute, Melbourne, Australia; [3]Department of Medical Biology, University of Melbourne, Melbourne, Australia; [4]Spatial Ecology and Epidemiology Group, Department of Zoology, University of Oxford, Oxford, United Kingdom; [5]Fogarty International Center, National Institutes of Health, Bethesda, United States; [6]Centre de Recerca en Salut Internacional de Barcelona, Barcelona, Spain

**Abstract** *Plasmodium vivax* relapse infections occur following activation of latent liver-stages parasites (hypnozoites) causing new blood-stage infections weeks to months after the initial infection. We develop a within-host mathematical model of liver-stage hypnozoites, and validate it against data from tropical strains of *P. vivax*. The within-host model is embedded in a *P. vivax* transmission model to demonstrate the build-up of the hypnozoite reservoir following new infections and its depletion through hypnozoite activation and death. The hypnozoite reservoir is predicted to be over-dispersed with many individuals having few or no hypnozoites, and some having intensely infected livers. Individuals with more hypnozoites are predicted to experience more relapses and contribute more to onwards *P. vivax* transmission. Incorporating hypnozoite killing drugs such as primaquine into first-line treatment regimens is predicted to cause substantial reductions in *P. vivax* transmission as individuals with the most hypnozoites are more likely to relapse and be targeted for treatment.

**\*For correspondence:** m.white08@imperial.ac.uk

## Introduction

The study of the transmission dynamics of vector-borne diseases such as *Plasmodium falciparum* malaria has a rich history, with a theoretical foundation based on the Ross-Macdonald models (*malERA Consultative Group on Modeling, 2011*; *Smith et al., 2012*; *Reiner et al., 2013*; *Smith et al., 2014*), a class of mathematical models describing the transmission of a pathogen between human and vector hosts. In the case of *P. falciparum*, the parasite has a reservoir in both the human host and the *Anopheles* mosquito, with transmission occurring when a mosquito takes a blood meal from a human. Ross-Macdonald models have provided insights into the dynamics of *P. falciparum* transmission resulting in valuable guidance for historical and contemporary malaria control programmes, most notably the large reductions in transmission that are achievable if the lifespan of the mosquito is reduced through vector control (*Macdonald, 1952a*; *Macdonald, 1952b*).

In contrast to the extensive theory of the mathematical epidemiology of *P. falciparum* malaria (*Smith et al., 2012*), *P. vivax* malaria has been comparatively neglected. This is in spite of *P. vivax* being the geographically most widely distributed species of malaria in the world, causing in the region of 80–300 million clinical episodes every year (*Mueller et al., 2009a*; *Gething et al., 2012*). *P. falciparum*

**eLife digest** Malaria is one of the world's most deadly infections, causing 100s of 1000s of deaths each year despite being both preventable and curable. Malaria is caused by *Plasmodium* parasites, which are transmitted between humans by mosquitoes. When a mosquito bites a human, *Plasmodium* is injected into the bloodstream with the mosquito's saliva. The parasite then travels through the bloodstream to the liver, infects liver cells and multiplies within those cells without causing any noticeable symptoms. After remaining silent in the liver for weeks or months, the now abundant parasite ruptures the host liver cell, re-enters the bloodstream, and begins infecting red blood cells. If another mosquito bites the infected individual and takes a blood meal, the parasite moves into the mosquito and the cycle of transmission continues.

There are several species of *Plasmodium* that are known to cause malaria. The most widely studied species is *P. falciparum*, which also causes one of the deadliest types of malaria. However, another *Plasmodium* species called *P. vivax* is the most widely distributed species and, despite being less virulent than *P. falciparum*, is particularly dangerous because it causes recurring malaria.

In contrast to *P. falciparum*, *P. vivax* has the ability to form hypnozoites: a dormant form of the parasite that can remain inside liver cells for long periods of time, sometimes for years. The reservoir of *P. vivax* hypnozoites can regularly populate the bloodstream with the infectious form of the parasite, triggering relapses of malaria. Even if an individual suffering a relapse receives prompt treatment to clear parasites in the blood, more parasites may emerge from the liver and cause new blood-stage infections.

White et al. developed a mathematical model to help understand how *P. vivax* is transmitted. Unlike many of the established models of malaria transmission, the new model accounts for the reservoir of *P. vivax* hypnozoites in the liver, and assumes that hypnozoites in the reservoir either die, or are activated and enter the bloodstream, at a constant rate. This produces patterns that closely match how often relapses occur in patients. White et al. go on to predict that although many infected people have few or no hypnozoites in their liver, some have many hypnozoites, and these people are more likely to suffer from malaria relapses. This suggests that if the initial treatments given to malaria sufferers incorporate additional drugs that kill the hypnozoites in the liver, then it may be possible to substantially reduce the extent of *P. vivax* transmission.

models are not applicable to *P. vivax* as they fail to account for the reservoir of dormant liver stages (hypnozoites) which give rise to relapsing infections—one of the defining characteristics of the biology and epidemiology of *P. vivax* (*Mueller et al., 2009a*). In natural transmission settings relapses are often undistinguishable from re-infections from new mosquito bites or recrudescences of existing blood-stage infections. When the origin of renewed parasitemia following primary *P. vivax* infection is unknown, it can be classified as a recurrent infection (*Battle et al., 2014*).

Most existing models of malaria transmission do not account for the additional reservoir of parasites in the liver, but the hypnozoite reservoir has been incorporated into some *P. vivax* transmission models as a state to denote hypnozoite infection (*Ishikawa et al., 2003*; *Chamchod and Beier, 2013*; *Roy et al., 2013*) or as up to two broods of hypnozoites (*Dezoysa et al., 1991*). Relapse patterns and their implications for transmission have also been investigated using statistical distributions for the time to first relapse (*Lover et al., 2014*). Here we advance on existing work by considering how the number of hypnozoites in the liver contributes to patterns of relapse infections and the epidemiology and control of *P. vivax*.

When a *P. vivax* infected mosquito takes a blood meal from a human, sporozoites are injected into the skin and migrate to the liver, where they invade hepatocytes and develop into either actively dividing schizonts or dormant hypnozoites. The development of actively dividing schizonts may lead to a primary blood-stage infection and potentially clinical malaria (*Mueller et al., 2013*). Hypnozoites will lie dormant in the liver for weeks to years before activating to initiate new blood-stage infections. The biological mechanisms regulating the activation of hypnozoites remain unknown (*Mueller et al., 2009a*), although a number of triggers for relapses have been proposed, including fever caused by other pathogens such as *P. falciparum* (*Shanks and White, 2013*) and exposure to *Anopheles*-specific proteins (*Hulden and Hulden, 2011*).

There is considerable geographical variation in the timing and frequency of *P. vivax* relapse infections, with strains from tropical areas having an average time to first relapse of 3–6 weeks and long-latency strains from temperate areas relapsing within 6–9 months (*Lover and Coker, 2013*; *Battle et al., 2014*). Beyond the first relapse, periodic patterns in multiple relapses from a single mosquito bite have been observed (*White, 2011*). For example, following a single infection with a tropical strain of *P. vivax* the time until next relapse has been observed to increase with each successive relapse (*Berliner et al., 1948*; *White, 2011*). In contrast, temperate strains are associated with a long latency period until first relapse (of the order of 6 months) followed by short intervals between successive relapses (*Coatney et al., 1950*; *Hankey et al., 1953*; *White, 2011*). A descriptive epidemiology of *P. vivax* relapses will thus require estimation of three key quantities: (i) the time to first relapse, (ii) the number of relapses per primary infection, and (iii) the duration of hypnozoite carriage.

In this manuscript, a within-host model of hypnozoites in liver hepatocytes is developed to demonstrate that many of the epidemiological patterns of relapse infections can be explained by making the assumption that hypnozoites activate and die at a constant rate. This model is integrated into the existing theory of Ross-Macdonald models to account for the relapse infections characteristic of *P. vivax* malaria. We use this model to provide qualitative insights into the relative contribution of relapses to *P. vivax* transmission and illustrate the consequences for controlling *P. vivax* with vector control and anti-malarial drugs.

## Results

### Within-host relapse model

*Figure 1* shows the best fit within-host relapse model to data on time to first relapse infection from three ecological zones with tropical strains of *P. vivax*: South America, South East Asia and Melanesia (*Battle et al., 2014*). In each ecological zone, the number of hypnozoites $N$ and the hypnozoite activation rate $\alpha$ were correlated (see *Figure 1—figure supplement 1*). For example, a short time to relapse could be explained by a single fast activating hypnozoite or a large number of slow activating hypnozoites. Longitudinal data where multiple relapses are observed in individuals would allow better estimation of the number of hypnozoites in the liver and the duration of hypnozoite carriage.

The within-host model can be used to simulate beyond the first relapse infection. *Figure 2* shows some sample relapse patterns from the within-host models for tropical and temperate strains of *P. vivax*. This model predicts notable dose-dependency with increased numbers of hypnozoites associated with a greater number of relapses and shorter time to first relapse. Following the long latency to first relapse in temperate strains, the interval between subsequent relapses is considerably shorter. The within-host model assumes that hypnozoites act independently of each other, and hence the time to next relapsing hypnozoite is exponentially distributed. In particular we do not predict periodicity between relapsing hypnozoites (in the absence of external triggers [*White, 2011*; *Shanks and White, 2013*]). If the simulated data are censored such that relapses occurring within 14 days of a previous relapse remain undetected (due to either prophylaxis by blood-stage anti-malarials or the presence of parasites from an existing infection) then there is an apparent periodicity in detected relapses. The observed periodicity of relapses will be determined by the duration of prophylactic protection and not via the biological mechanisms considered here. The periodicity in detected relapses is most evident for large numbers of hypnozoites with the period being determined by the assumed duration of prophylactic protection (*Figure 2—figure supplement 1*). However, as has been previously argued, periodicity in relapses could also be attributable to a cycle of fevers initiating hypnozoite activation which in turn cause new blood-stage infections and malaria-associated fevers (*White, 2011*).

*Figure 3* shows the predicted number of relapsing hypnozoites in a population exposed to *P. vivax* in the absence of new infections. For tropical strains the mean number of hypnozoites in the liver is expected to decrease exponentially, but the proportion of individuals carrying hypnozoites is expected to decrease at a slower rate as an individual can relapse even if they have just one hypnozoite (*Figure 3A*). For temperate strains the mean number of hypnozoites in the liver decreases slowly, as hypnozoites remain in the long-latency phase for approximately 6 months (*Figure 3B*). The model allows estimation of time to second, third and consecutive relapses in addition to estimates of time to first relapse

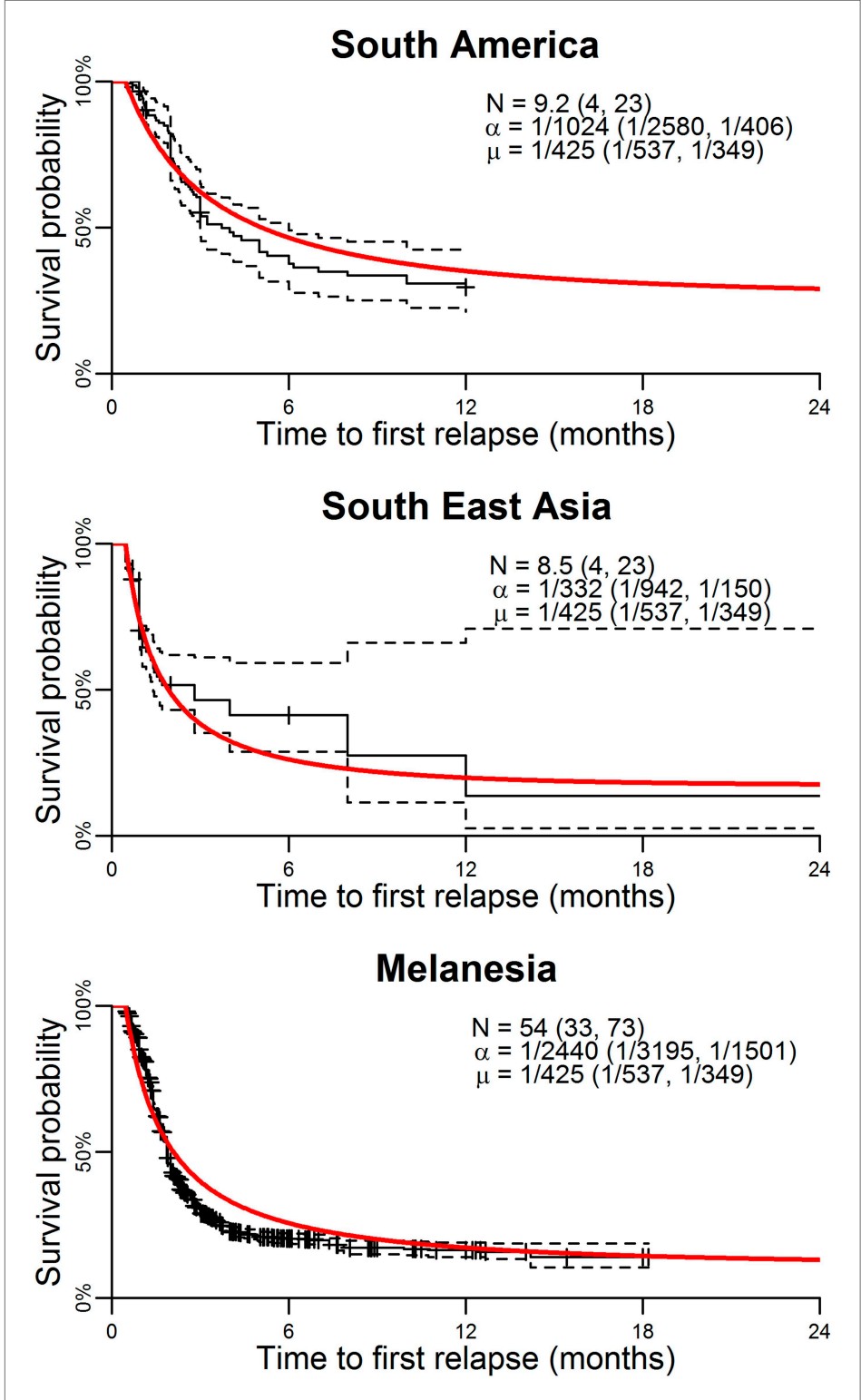

**Figure 1**. Model parameterisation. Time to first relapse infection from the within-host model fitted to data from three ecological zones with tropical strains of *P. vivax* (***Battle et al., 2014***). The red curves show the model fits with estimated posterior median parameters.

The following figure supplement is available for figure 1:

**Figure supplement 1**. MCMC chains and posterior distributions for Bayesian model fitting.

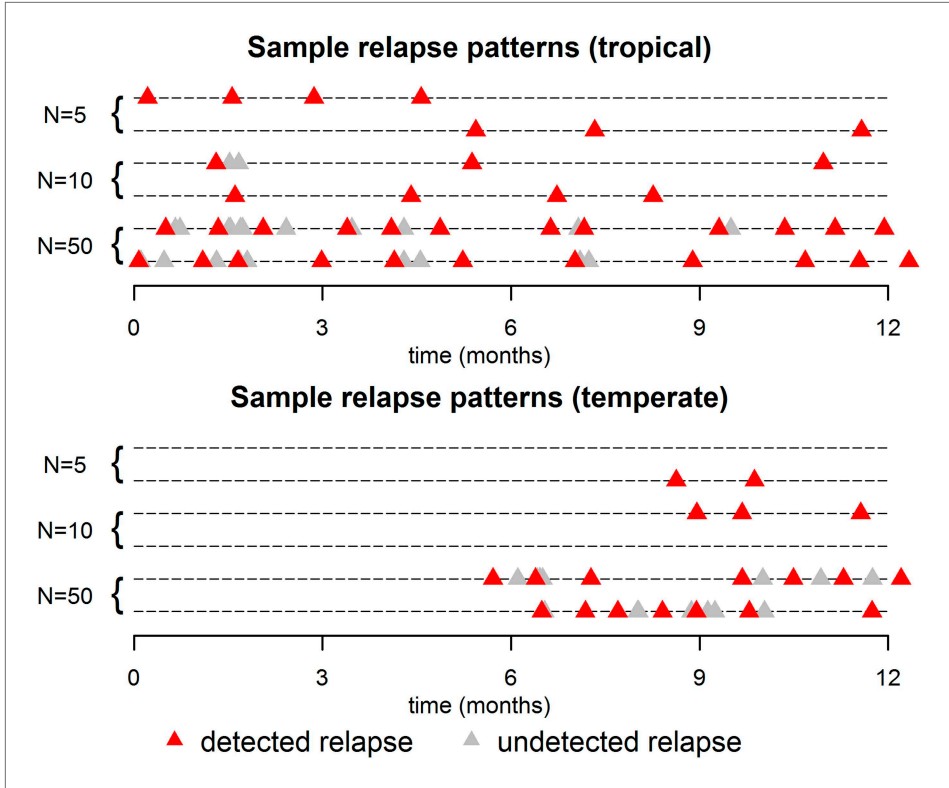

**Figure 2**. Sample relapse patterns for tropical and temperate strains of *P. vivax*. A relapse is assumed to be undetected if it occurs within 14 days of a detected relapse. Both tropical and temperate phenotypes exhibit dose dependency, with a larger number of hypnozoites giving rise to a greater number of relapses and shorter times to first relapse. For larger numbers of hypnozoites (*N* = 50), periodicity in detected relapses is observed. The appearance of this periodicity is due to the undetected relapses.

The following figure supplement is available for figure 1:

**Figure supplement 1**. Expected time between consecutive relapses.

obtainable via survival analysis of patient data (*Lover and Coker, 2013*; *Battle et al., 2014*) (*Figure 3C,D*). The expected number of relapsing hypnozoites per individual is expected to follow an approximately exponential distribution (*Figure 3E,F*) in agreement with empirical observations (*Horing, 1947*; *White, 2011*).

## Dynamics and steady states of *P. vivax* transmission

*Figure 4A* shows the predicted steady states (the equilibrium blood-stage prevalence in the absence of seasonally varying transmission) as a function of entomological inoculation rate (EIR). EIR is a measurement of the number of infectious bites per person per year. The proportion of people infected with hypnozoites is predicted to be higher than the proportion infected with *P. vivax* blood-stage parasites. For a given EIR, *P. vivax* blood-stage prevalence is predicted to be higher than *P. falciparum* prevalence as a single mosquito bite can give rise to multiple blood-stage infections. However this does not account for the longer duration of *P. falciparum* infections as a consequence of antigenic switching (*Molineaux et al., 2001*), and the important role of heterogeneity in exposure (*Smith et al., 2005*). With the exception of the *P. vivax* hypnozoite rate, these quantities can be measured in epidemiological field studies (*Smith et al., 2005*; *Kelly-Hope and McKenzie, 2009*).

*Figure 4B* shows how the median number of hypnozoites increases with increasing *P. vivax* transmission. *Figure 4C* shows the distribution in the number of hypnozoites when *PvPR* = 50%. The

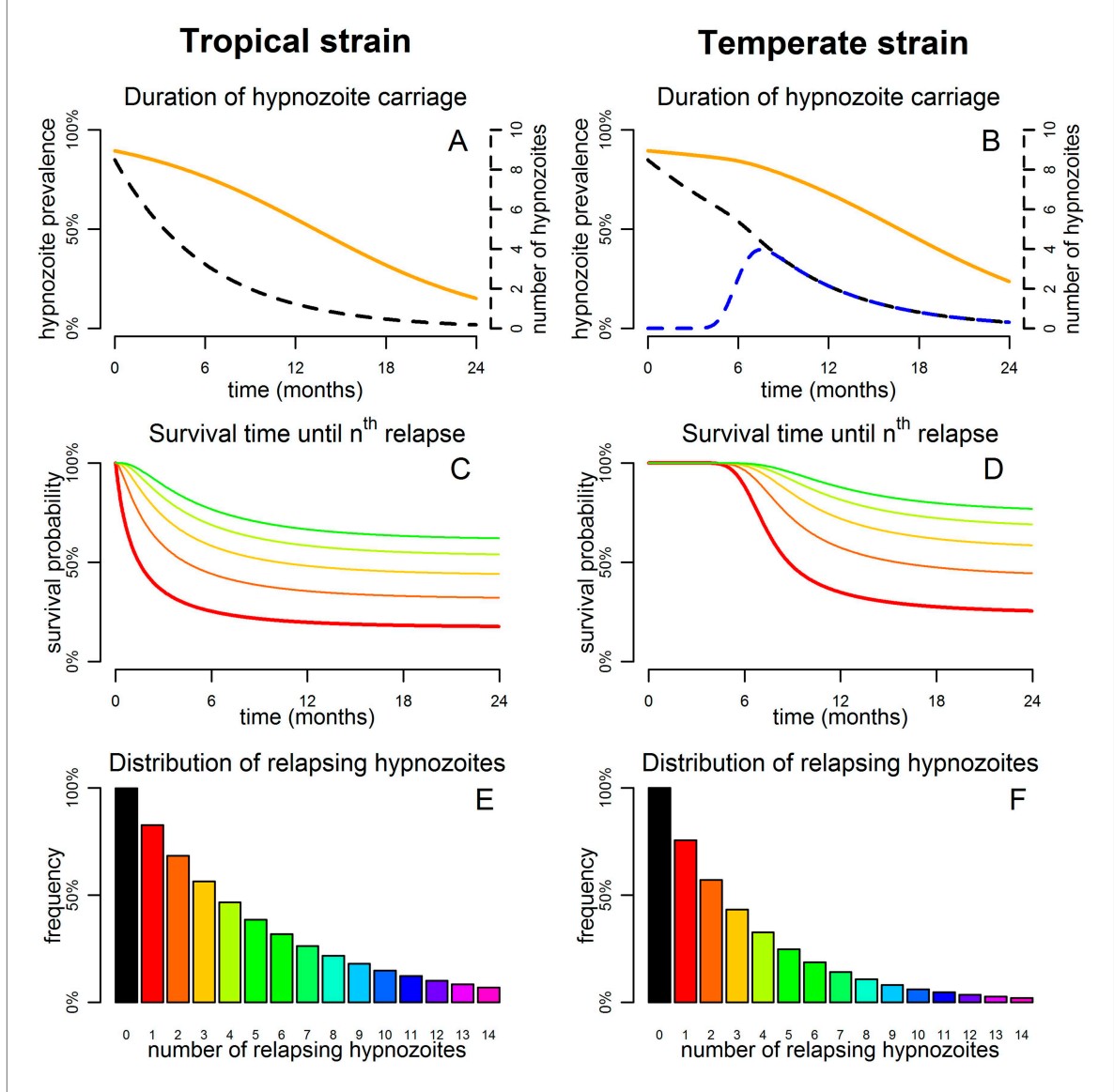

**Figure 3**. Predicted relapse infections following primary *P. vivax* infection. (**A** and **B**) Duration of hypnozoite carriage (orange) and expected number of hypnozoites in the liver (dashed). For the temperate strain, the dashed blue line shows the number of hypnozoites in the relapsing phase. (**C** and **D**) Survival time until *n*th relapsing hypnozoite. The red curve is equivalent to the Kaplan–Meier curve for time to first blood-stage infection that would be observed in the absence of new infections from mosquito bites. Only the curves for the first five relapses are shown. (**E** and **F**) Proportion of individuals with at least *n* relapsing hypnozoites following primary infection.

number of hypnozoites per individual is predicted to be over-dispersed following a negative binomial distribution. Thus some individuals will harbour a large number of hypnozoites while some will have none. This phenomenon will be further amplified if there is heterogeneity in exposure where some individuals receive a large number of mosquito bites.

## Control of *P. vivax*

The impact of malaria control interventions will depend on how effectively the parasite is targeted in each of the reservoirs in the mosquito, the blood and the liver. *Figure 5* shows the qualitative effects of malaria control on the transmission dynamics of *P. falciparum* and *P. vivax*. Vector control with insecticide treated nets (ITNs) or indoor residual spraying (IRS) is assumed to increase mosquito mortality. The introduction of vector control is expected to cause a

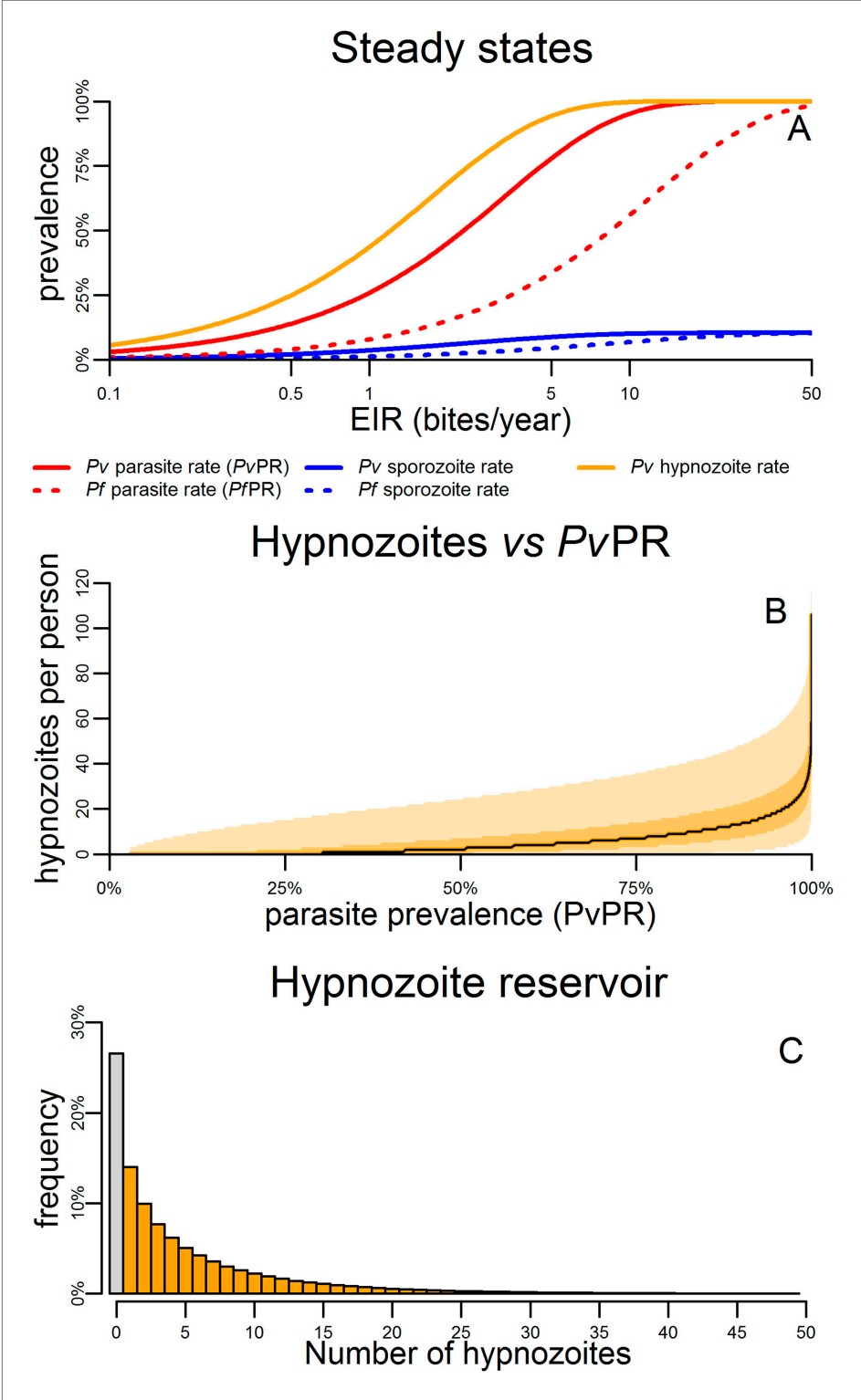

**Figure 4**. Within-host model for tropical relapses embedded in a *P. vivax* transmission model. (**A**) The statics (estimated equilibrium prevalence) of *P. vivax* and *P. falciparum* transmission for different values of the entomological inoculation rate (EIR). EIR was varied by changing the number of mosquitoes per person *m*. (**B**) The number of hypnozoites per person is expected to increase with transmission intensity. The black line denotes the median number of hypnozoites, and the shaded areas denote the 50% and 95% ranges. (**C**) The distribution of the hypnozoite reservoir when *Pv*PR = 50%. The grey bar represents individuals with zero hypnozoites.

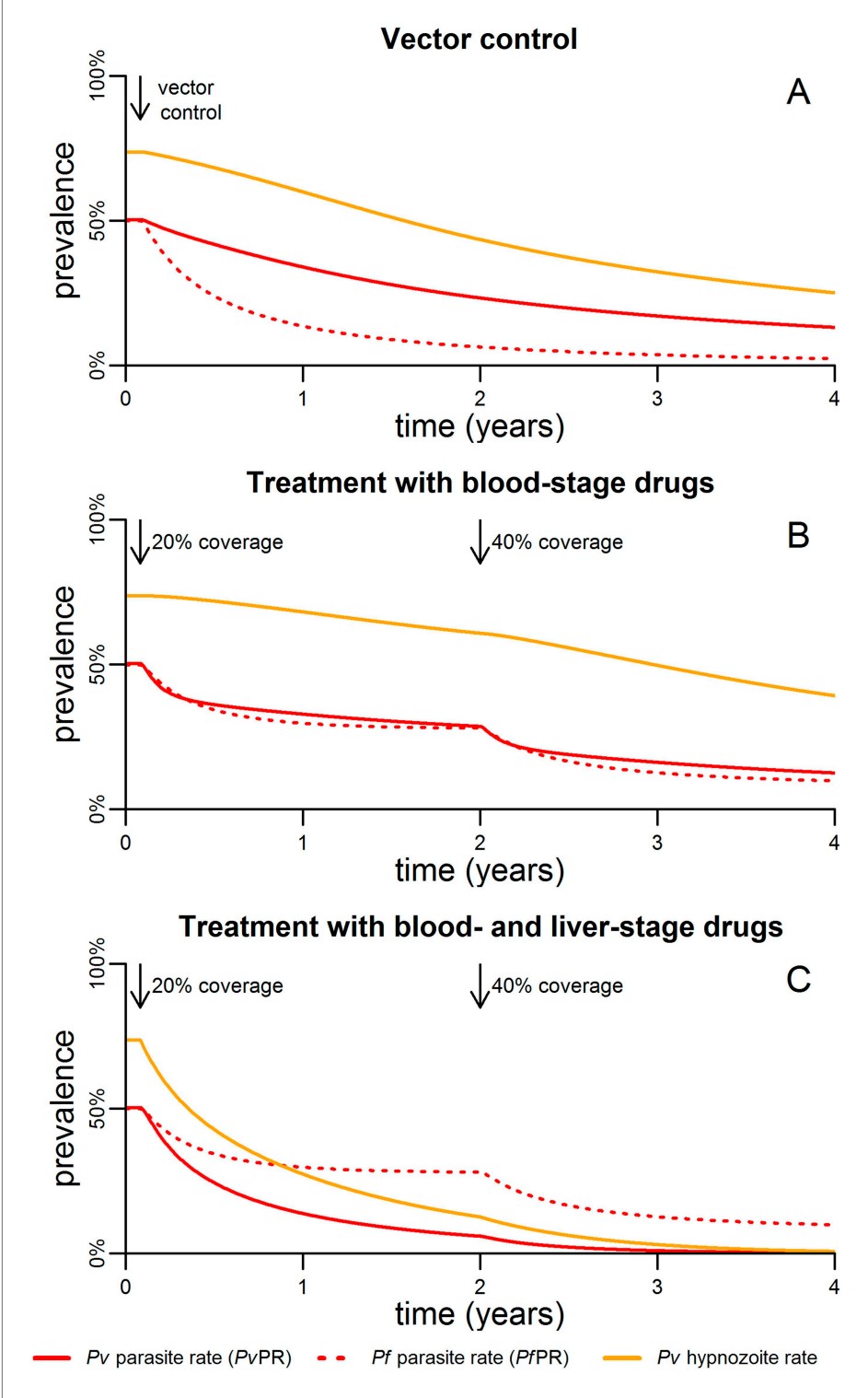

**Figure 5**. Timelines for malaria control. (**A**) The introduction of vector control with ITNs or IRS (assumed to increase mosquito mortality by 30%) is predicted to cause substantial reductions in both *Pv*PR and *Pf*PR. (**B**) Simulated effect of expanding first-line treatment with blood-stage anti-malarial drugs (e.g., chloroquine or ACTs) so that 20% and 40% of new blood-stage infections are treated. (**C**) Simulated effect of first-line treatment with a combined regimen of blood-stage anti-malarials and primaquine to remove liver-stage hypnozoites.

*Figure 5. Continued on next page*

*Figure 5. Continued*

The following figure supplements are available for figure 5:

**Figure supplement 1**. Transmission model incorporating treatment of new infections with blood-stage anti-malarials.

**Figure supplement 2**. Transmission model incorporating treatment of new infections with blood-stage anti-malarials and primaquine.

rapid decline in *P. falciparum* parasite rate (*Pf*PR), and a smaller and slower decline in *P. vivax* (*Figure 5A*).

*Figure 5B* shows the effect of targeting the parasite reservoir in the blood by providing first-line treatment for new blood-stage infections with anti-malarial drugs such as chloroquine or artemisinin combination therapies (ACTs). See details of how treatment was implemented in the model are provided in *Figure 5—figure supplement 1,2*. Increasing treatment coverage leads to reductions in blood-stage prevalence of both *P. falciparum* and *P. vivax*. Notably the reduction in the *P. vivax* hypnozoite rate is slow as the hypnozoite reservoir is not directly targeted.

The hypnozoite reservoir can be directly targeted using a drug such as primaquine that can eliminate hypnozoites from the liver (*Wells et al., 2010*). The inclusion of primaquine in first-line treatment regimens is predicted to cause substantial reductions in both the *P. vivax* parasite rate and hypnozoite rate (*Figure 5C*), as individuals being treated for blood-stage *P. vivax* infections will also have their hypnozoites removed. A consequence of this strategy is that the hypnozoite reservoir can be targeted efficiently, as individuals with the most hypnozoites are most likely to relapse and potentially be detected by health systems. *Figure 6* shows how the inclusion of primaquine in first-line treatment regimens preferentially targets the most intense infections, with the greatest reductions observed in individuals with the most hypnozoites.

## Discussion

Relapse infections arising from the activation of hypnozoites in the human liver have important consequences for the transmission dynamics of *P. vivax*. Hypnozoites in the liver constitute a third malaria parasite reservoir, in addition to the reservoirs in the blood circulation and mosquito also present for *P. falciparum*. Relapses can be incorporated into Ross-Macdonald models of malaria transmission through the addition of a state to represent the hypnozoite reservoir (*Roy et al., 2013*), or as demonstrated here, through consideration of the number of hypnozoites in the liver. This allows the intensity of hypnozoite infection to be estimated which is crucial for understanding patterns of relapse infections (*White, 2011*) and evaluating the effect of interventions such as primaquine treatment that directly target the hypnozoite reservoir. Hypnozoites are assumed to be subjected to two processes: activation at constant rate α, and death at constant rate μ. By considering infection with batches of hypnozoites, these simple processes can explain many of the complex patterns observed in *P. vivax* relapses (see *Box 1*).

A key assumption in the proposed model is the constant activation of hypnozoites. This implies that relapses can occur immediately after primary *P. vivax* infection, in contrast to suggestions that the first relapse does not occur until 2–3 weeks later. The best evidence on early relapses comes from treatment efficacy studies where patients treated for *P. vivax* are followed for recurrent infection for 42 days (*Douglas et al., 2010*). Except in cases with documented chloroquine resistance (*Price et al., 2014*), recurrent infections are rarely observed prior to day 14, however the inclusion of long-lasting anti-malarials in treatment regimens provides prophylaxis during this period making detection of parasites unlikely. In a recent study of the slowly eliminated drug dihydroartemisinin-piperaquine (DP) *Tarning et al. (2014)* tested a model where relapses occur in bursts every 3 weeks, but it arguably provided no better fit to the data than a model of constant hypnozoite activation. Testing the hypothesis of constant activation would require follow up of patients treated with rapidly eliminated artemisinin monotherapy, a challenging proposition given the concerns over artemisinin resistant *P. falciparum* (*Ashley et al., 2014*).

Although the model captures the key drivers of the dynamics of *P. vivax* transmission, it is a simplified representation subject to a number of limiting assumptions. The potential role of triggers of

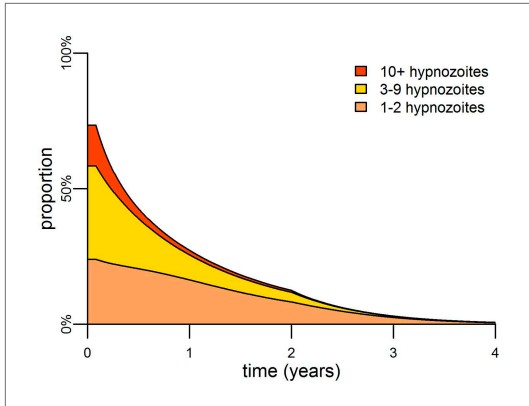

**Figure 6**. Targeting the hypnozoite reservoir. Proportion of the population infected with 1–2, 3–9 or 10+ hypnozoites following the introduction of a first-line treatment regimen with blood-stage anti-malarial drugs and primaquine. Individuals with large numbers of hypnozoites are more likely to experience new blood-stage infections and hence become targeted for treatment and have their hypnozoites removed. This results in a selective targeting of the most intensely infected individuals.

hypnozoite activation such as febrile illness (**Shanks and White, 2013**) are not accounted for. There is no heterogeneity or seasonality in transmission, and no age structure. Incorporation of the acquisition of natural immunity into the model will be particularly important for settings with high transmission intensity where immunity has a role in regulating blood-stage infections (**Mueller et al., 2013**). It is assumed that all individuals infected with blood-stage parasites are capable of transmitting to mosquitoes. Similar to the corresponding *P. falciparum* models, incorporation of these factors would change the quantitative predictions of the model, but not its qualitative behaviour.

*P. falciparum* and *P. vivax* parasite prevalence (*Pf*PR and *Pv*PR) are the most widely reported and best validated metrics of malaria transmission from epidemiological studies (**Gething et al., 2011a**; **Gething et al., 2012**) providing measurements of the proportion of individuals with detectable blood-stage parasites. In settings with similar levels of *P. vivax* and *P. falciparum* transmission, the model predicts *Pv*PR to be greater than *Pf*PR due to the additional blood-stage infections arising from relapses. However, this does not agree with empirical observations which find *Pf*PR to be similar to or greater than *Pv*PR (**Snounou and White, 2004**; **Mueller et al., 2009b**). This is most likely explained by the rapid acquisition of immunity to *P. vivax* (**Koepfli et al., 2013**; **Mueller et al., 2013**) the low detectability of *P.* vivax blood-stage infections (**Harris et al., 2010**), and the longer durations of *P. falciparum* blood-stage infection (**Molineaux et al., 2001**) not captured in the model. Furthermore, the additional *P. vivax* parasite reservoir in the liver means that *Pf*PR and *Pv*PR are not directly comparable metrics. Thus if a parasitological survey indicates similar parasite prevalence, a greater control effort will be required to reduce *P. vivax* transmission than to reduce *P. falciparum* transmission because of the additional infections emerging from the hypnozoite reservoir. The model described here allows the proportion of individuals harbouring hypnozoites to be estimated given metrics such as *Pv*PR. The number of hypnozoites per person is predicted to be over-dispersed with some individuals with intensely infected livers and most carrying few or no hypnozoites (**Figure 4C**). Estimates of the prevalence and intensity of hypnozoite infection will be dependent on both the uncertainty in the measurable data and the model assumptions.

Vector control with ITNs or IRS, and treatment with effective anti-malarial drugs are the cornerstones of malaria control efforts targeting the parasite in the vector and the human host, however they are predicted to have different effects on *P. vivax* and *P. falciparum* transmission. Vector control interventions that increase mosquito mortality are expected to cause greater reductions in *Pf*PR than *Pv*PR (**Figure 5A**), as higher levels of *P. vivax* transmission can be maintained with fewer mosquito bites. This has been observed in both Thailand (**Sattabongkot et al., 2004**) and Brazil (**Coura et al., 2006**) where increased vector control has caused greater reductions in *P. falciparum* than *P. vivax*.

First-line treatment of new blood-stage infections with anti-malarial drugs such as chloroquine or ACTs is predicted to cause moderate reductions in blood-stage prevalence of both *P. falciparum* and *P. vivax* (**Figure 5B**). The addition of primaquine to first-line treatment regimens is expected to cause large reductions in *P. vivax* blood-stage prevalence, as individuals with the most intense hypnozoite infections are more likely to relapse and be targeted for treatment and hence have their hypnozoites eliminated. The potential to simultaneously target parasite reservoirs in the blood and liver may turn the cause of *P. vivax* parasites' robust transmission into its Achilles' heel.

In *P. vivax* and *P. falciparum* co-endemic regions, heterogeneity in exposure to mosquito bites may cause associations between *P. falciparum* fevers and the risk of future *P. vivax* relapses

**Box 1.** Properties of predicted *P. vivax* relapse patterns

Nicholas White (**White, 2011**) has argued that a theory seeking to explain the remarkable biology of *P. vivax* relapses must accommodate eight phenomena. The within-host relapse model proposed here accounts for six of these, and can be extended to accommodate the other two.

- Relapses show remarkable periodicity. The assumption of constant hypnozoite activation and death does not allow for periodic relapses. However if we assume some relapses are undetected due to prophylaxis following treatment or the presence of existing blood-stage parasites, an apparent periodicity in detected relapses is predicted (**Figure 2**).
- For tropical strains of *P.* vivax the highest incidence of relapses is predicted to be immediately after primary infection (**Figure 1**), thus the first relapses will be detected after the prophylactic effects of treatment have waned.
- Not all *P. vivax* infections are followed by relapses. Individuals will avoid relapses if the number of initial hypnozoites is zero, or if hypnozoites die before relapsing (**Figure 3D**).
- Multiple relapses are common. The within-host model predicts an approximately exponential distribution in the numbers of relapsing hypnozoites per person (**Figure 3E,F**). Dose-dependency in the number of relapses is predicted, with greater numbers of hypnozoites giving rise to greater numbers of relapses.
- The time to first relapse for temperate strains is predicted to follow a gamma distribution (approximately Normal).
- After the first relapse of a temperate strain, the intervals between consecutive relapses are predicted to be similar to those observed in tropical strains (**Figure 2**).

The model described here does not account for the varying genotypes arising from single or multiple mosquito bites, although it can be extended to account for the diversity of genotypes in the hypnozoite reservoir (**Koepfli et al., 2011**, **2013**). The model represents a baseline scenario with constant hypnozoite activation and death in the absence of external triggers. The role of triggers such as fever can be tested for in detail with data on the timing and peak temperatures of episodes of febrile illness.

(**Douglas et al., 2011**). Thus the inclusion of primaquine in first-line treatment for *P. falciparum* may also reduce *P. vivax* transmission. In addition to inclusion in first-line treatment regimens, primaquine can also be administered as part of mass drug administration (MDA) programmes. In treatment-reinfection studies of Papua New Guinean children (**Betuela et al., 2012**; Robinson et al., unpublished), mass administration of drugs such as chloroquine or artemether-lumefantrine successfully cleared *P. vivax* blood-stage infections but rapid recurrence of infection was observed during follow-up—most likely due to relapses. The addition of primaquine to the treatment regimen caused large reductions in the rate of recurrent infections.

Although primaquine treatment clears the hypnozoite reservoir, it requires a difficult 14 day treatment regimen, and is not without risk due to vulnerability to haemolytic toxicity among glucose-6-phosphate dehydrogenase (G6PD) deficient patients (**Howes et al., 2013**). Individuals should thus be tested for G6PD deficiency (**Kim et al., 2011**) before the administration of primaquine. A primaquine analogue, tafenoquine, is currently undergoing phase three trials and is likely to be licensed for use by 2017 (**Llanos-Cuentas et al., 2014**). Tafenoquine requires a single dose alongside a 3 day chloroquine regimen, but is subject to the same risks in G6PD deficient patients. A quantitative model of *P. vivax* transmission will allow for the benefits of primaquine treatment to be weighed against the risks of G6PD deficiency and the costs of G6PD testing.

Mathematical models of malaria transmission that account for *P. vivax* relapses can provide valuable insights into the impact of malaria control interventions on the parasite's reservoirs in the vector, the blood and the liver. In the absence of effective diagnostics for detecting liver-stage parasites (**malERA Consultative Group on Diagnoses and Diagnostics, 2011**), models will play a crucial role in estimating and predicting the effectiveness of interventions that target the hypnozoite reservoir, either indirectly via vector control and blood-stage anti-malarials or directly via primaquine treatment.

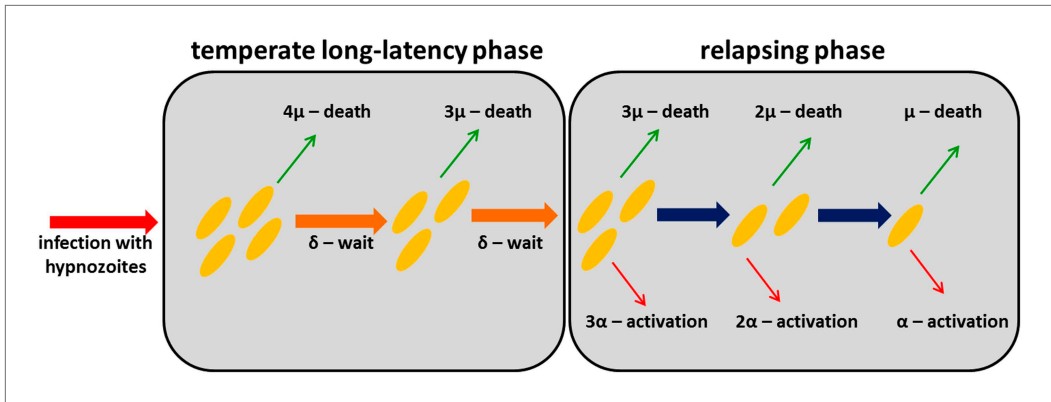

**Figure 7**. Within-host model schematic of relapsing hypnozoites in the liver. Hypnozoites from tropical strains of *P. vivax* will progress to the relapsing phase where they are subject to two processes: death and activation leading to relapse. Hypnozoites from temperate strains will begin in a temperate long-latency phase where they must wait before progressing to the relapsing phase.

The following figure supplement is available for figure 7:

**Figure supplement 1**. Detailed model schematic of the within-host relapse model.

## Materials and methods

### Within-host model for tropical relapses

Following infection with a tropical strain of *P. vivax*, the population dynamics of hypnozoites in the liver can be described by a within-host model where each hypnozoite is subject to two processes: (i) activation leading to relapse infection; and (ii) death, either of the hypnozoite itself or the host hepatocyte (*Malato et al., 2011*). Constant activation (α) and death (μ) rates are assumed implying hypnozoite residence time in the liver is exponentially distributed. The long latency of temperate strains before first relapse can be accounted for by assuming a period of dormancy during which hypnozoites must wait before they can activate. A schematic representation of the within-host model is presented in *Figure 7*.

The tropical relapse model is assumed to begin with an initial population of $N$ hypnozoites, each of which can either activate at rate α, or die at rate μ, independently of each other. The number of hypnozoites in the liver will decay exponentially with an expected $Ne^{-(\mu+\alpha)t}$ hypnozoites at time $t$. Let $H_i^N(t)$ denote the probability that $i$ of $N$ hypnozoites remain in the liver after time $t$. The hypnozoite population dynamics can be described by the following set of equations:

$$\frac{dH_N^N}{dt} = -(\mu+\alpha)NH_N^N$$

$$\frac{dH_i^N}{dt} = -(\mu+\alpha)iH_i^N + (\mu+\alpha)(i+1)H_{i+1}^N, \quad i = 0...N-1 \tag{1}$$

*Equation 1* can be solved analytically to give:

$$H_i^N(t) = \binom{N}{i} e^{-N(\mu+\alpha)t} \left(e^{(\mu+\alpha)t} - 1\right)^{N-i} \tag{2}$$

Define $P_j^N(t)$ to be the probability that $j$ relapses have occurred by time $t$. This can be calculated as follows: if $i$ hypnozoites remain in the liver, then $N - i$ have either activated or died. The probability of each hypnozoite activating is $\frac{\alpha}{\mu+\alpha}$. The probability that $j$ of $N - i$ hypnozoites have activated can thus be calculated from a binomial distribution. Summing over the allowable number of hypnozoites (at least $j$ hypnozoites must have activated or died for $j$ relapses to be observed) gives:

$$P_j^N(t) = \sum_{i=0}^{N-j} \binom{N-i}{j} \left(\frac{\alpha}{\mu+\alpha}\right)^j \left(\frac{\mu}{\mu+\alpha}\right)^{N-i-j} H_i^N(t)$$

$$= \binom{N}{j} \frac{\alpha^j \mu^{N-j}}{(\mu+\alpha)^N} \left(1 - e^{-(\mu+\alpha)t}\right)^N \left(1 + \frac{\mu+\alpha}{\mu} \frac{1}{e^{(\mu+\alpha)t}-1}\right)^{N-j} \quad (3)$$

*Equations 1–3* describe the population dynamics of hypnozoites in a single individual with *N* hypnozoites in the absence of exposure to new infections. In a population of individuals, we would expect substantial variation in the numbers of hypnozoites due to heterogeneity in exposure and the variation in sporozoite inoculum from each infectious mosquito bite (*Beier et al., 1991*; *Medica and Sinnis, 2005*; *White et al., 2013*). Based on evidence that the number of sporozoites injected with a mosquito bite approximately follows a geometric distribution (*Beier et al., 1991*), we assume that the number of hypnozoites following a primary infection is also geometrically distributed. If the mean number of hypnozoites is *N*, then the probability of *k* hypnozoites is $\left(\frac{N}{N+1}\right)^k \frac{1}{N+1}$. Assuming a geometrically distributed number of hypnozoites, the three quantities describing the epidemiology of relapses can be estimated in terms of the within-host parameters. The expected number of relapsing hypnozoites is:

$$h = \sum_{k=0}^{\infty} \underbrace{\left(\frac{N}{N+1}\right)^k \frac{1}{N+1}}_{\substack{\text{probability of } k \\ \text{initial hypnozoites}}} k \underbrace{\frac{\alpha}{\alpha+\mu}}_{\substack{\text{probability of each} \\ \text{hypnozoite relapsing}}} = N\frac{\alpha}{\mu+\alpha} \quad (4)$$

The mean duration of hypnozoite carriage is:

$$\frac{1}{\gamma} = \sum_{k=0}^{\infty} \underbrace{\left(\frac{N}{N+1}\right)^k \frac{1}{N+1}}_{\substack{\text{probability of } k \\ \text{initial hypnozoites}}} \underbrace{\sum_{i=1}^{k} \frac{1}{i} \frac{1}{\mu+\alpha}}_{\substack{\text{duration of} \\ k \text{ hypnozoites}}} = \frac{\log(N+1)}{\mu+\alpha} \quad (5)$$

The expected time to first relapse is:

$$\frac{1}{f} = \sum_{k=1}^{\infty} \underbrace{\left[\left(\frac{N}{N+1}\right)^k \frac{1}{N}\right]}_{\substack{\text{probability of } k \\ \text{initial hypnozoites}}} \sum_{i=1}^{k} \underbrace{\left[\left(\frac{\mu}{\mu+\alpha}\right)^{k-i} \frac{\alpha}{\mu+\alpha} \bigg/ \left(1 - \left(\frac{\mu}{\mu+\alpha}\right)^k\right)\right]}_{\text{probability of hypnozoite i being first to relapse}} \underbrace{\sum_{j=0}^{k-i} \frac{1}{k-j} \frac{1}{\mu+\alpha}}_{\text{time to relapse}} \quad (6)$$

The within-host relapse model describes a baseline scenario in the absence of potential external triggers for relapse such as fever (*Shanks and White, 2013*). Underlying assumptions of this model are: (i) each hypnozoite acts independently of other hypnozoites, for example, hypnozoites will not activate in batches due to mechanisms such as quorum sensing; and, (ii) hypnozoite death occurs at a constant rate, due to either death of the hypnozoite within the hepatocyte or death of the hepatocyte itself (*Malato et al., 2011*). The activation of a hypnozoite may not directly correspond to a detected relapse. For example, an infection arising from two hypnozoites activating within a day of each other is likely to be classified as a single relapse.

## Within-host model for temperate relapses

The within-host model can be extended to account for temperate strains of *P. vivax*. We assume that before a hypnozoite is capable of activating, it must undergo a long-latency phase of duration *d*. During this period hypnozoites are subject to death at rate µ. In particular, we assume that the time spent in the temperate long-latency phase can be described by a gamma distribution with mean *d* and variance $d^2/M$. This gamma distribution can be simulated by *M* successive compartments with exponential waiting times $1/\delta = d/M$. Increasing the number of compartments *M* reduces the variance in the duration of the dormancy period (*Wearing et al., 2005*). Following a primary infection where *N* hypnozoites of a temperate phenotype develop in the liver, we define $L_{i,j}^N$ as the probability that *i* of *N* hypnozoites are waiting in long-latency compartment number *j*, then the number of dormant and potentially active hypnozoites can be described by the following system of differential equations.

$$\frac{dL_{N,1}^N}{dt} = -\delta L_{N,1}^N - N\mu L_{N,1}^N$$

$$\frac{dL_{i,1}^N}{dt} = -\delta L_{i,1}^N - i\mu L_{i,1}^N + (i+1)\mu L_{i+1,1}^N \qquad i = 1...N-1$$

$$\frac{dL_{N,j}^N}{dt} = -\delta L_{N,j}^N + \delta L_{N,j+1}^N - N\mu L_{N,j}^N \qquad j = 2...M$$

$$\frac{dL_{i,j}^N}{dt} = -\delta L_{i,j}^N + \delta L_{i,j+1}^N - i\mu L_{i,j}^N + (i+1)\mu L_{i+1,j}^N \qquad i = 1...N-1, \ j = 2...M$$

$$\frac{dH_N^N}{dt} = \delta L_{N,M}^N - N(\mu + \alpha)H_N^N$$

$$\frac{dH_i^N}{dt} = \delta L_{i,M}^N - i(\mu + \alpha)H_i^N + (i+1)(\mu + \alpha)H_{i+1}^N \qquad i = 1...N-1$$

$$\frac{dH_0^N}{dt} = \sum_{j=1}^M \mu L_{1,j}^N + (\mu + \alpha)H_1^N \tag{7}$$

The equations are presented schematically in *Figure 7—figure supplement 1*. *Equation 7* cannot be solved analytically and must be computed numerically to calculate $H_i^N(t)$ and $L_{i,j}^N(t)$. A greater deal of uncertainty surrounds the biological processes accounting for the initial long-latency phase observed in temperate strains of *P. vivax*. In the model implemented here, it is assumed that all hypnozoites in an infection must undergo some waiting period before any of them can activate, and that during the long-latency phase hypnozoites are at risk of death due to natural hepatocyte death.

## *P. vivax* transmission model

We next embedded the within-host model for tropical relapses in a model for the transmission of *P. vivax* between humans and mosquitoes (*Figure 8*). The transmission dynamics are driven by two processes: (i) transmission of parasites through mosquito bites; and (ii) relapsing of liver-stage hypnozoites to cause new blood-stage infections. As per the standard Ross-Macdonald theory, the force of blood-stage infections in humans can be calculated as the product of the number of mosquitoes per human $m$, the rate at which each mosquito bites a human host $a$, the probability of transmission from mosquito to human following an infectious bite $b$, and the proportion of mosquitoes that are infectious $I_M$, to give $\lambda = mabI_M$. Parameter values are provided in *Table 1*. The force of infection on mosquitoes can be calculated in a similar manner. We assume that people can be susceptible ($S_i$) or infected with blood-stage parasites ($I_i$), where $i$ denotes the number of hypnozoites in the liver.

The increase in hypnozoites in the liver is determined by the force of infection $\lambda$ and the number of hypnozoites per infection $N$, and the decrease is due to hypnozoite activation $\alpha$ and death $\mu$. The model depicted in *Figure 8* can be described by the following set of equations:

$$\frac{dS_i}{dt} = -\lambda S_i - i(\mu + \alpha)S_i + (i+1)\mu S_{i+1} + \rho_i I_i \qquad i = 0...\infty$$

$$\frac{dI_i}{dt} = -\lambda I_i + \sum_{j=0}^i \lambda_{j \to i}\left(S_j + I_j\right) - i(\mu + \alpha)I_i$$

$$\qquad\qquad + (i+1)(\mu + \alpha)I_{i+1} + (i+1)\alpha S_{i+1} - \rho_i I_i \qquad i = 0...\infty$$

$$\frac{dS_M}{dt} = g - ac\left(\sum_{i=0}^\infty I_i\right)\left(e^{-gn} - I_M\right) - gS_M$$

$$\frac{dI_M}{dt} = ac\left(\sum_{i=0}^\infty I_i\right)\left(e^{-gn} - I_M\right) - gI_M \tag{8}$$

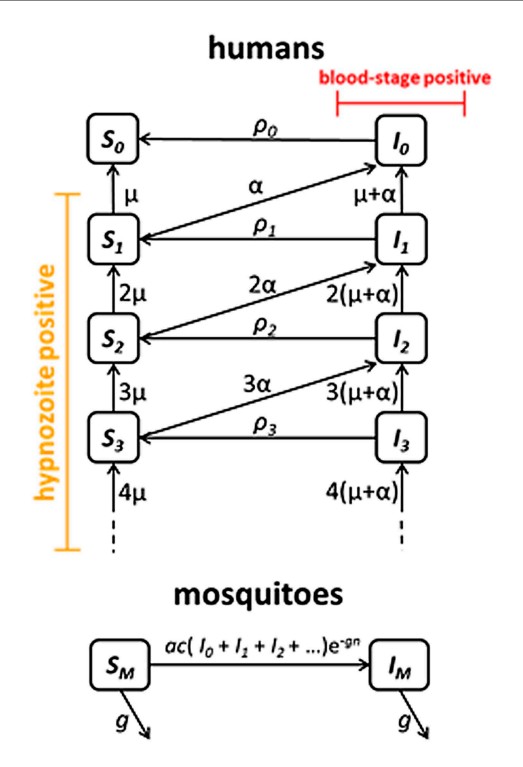

**humans**

**Figure 8**. Transmission model schematic. Within-host model for tropical relapses embedded in a transmission model. $S_i$ denotes the proportion of humans susceptible to blood-stage infection with $i$ hypnozoites. $I_i$ denotes the proportion of humans with blood-stage infections carrying $i$ hypnozoites. Individuals in all compartments are exposed to primary infections at rate $\lambda$, following which they will move down the flow diagram to a compartment representing blood-stage infection and carrying a greater number of hypnozoites.

where $\lambda_{j \to i} = \lambda \left( \dfrac{N}{N+1} \right)^{i-j} \dfrac{1}{N+1}$. In the absence of super-infection, the recovery from blood-stage infection is $\rho_i = r$. Accounting for super-infections (**Dietz and Molineaux, 1973**; **Smith et al., 2012**) gives $\rho_i = \dfrac{\lambda + i\alpha}{e^{\frac{\lambda+i\alpha}{r}} - 1}$.

## Model parameterisation

The within-host model for tropical relapses was fitted in a Bayesian framework to data on time to first relapse infection from three ecological zones with tropical strains of *P. vivax*: South America, South East Asia and Melanesia (see **Source data 1**). The data are described in detail by **Battle et al. (2014)**. Individual-level data on time to first recurrence was collated from individuals exposed to *P. vivax* infection (either via natural exposure or artificial challenge) and mostly followed up in the absence of exposure to new infections (**Battle et al., 2014**). The likelihood of the tropical relapse model can be evaluated by applying the model to the data on time to first relapse infection. The first detected relapse will occur after clearance of parasites from the primary infection and after the period of prophylactic protection from anti-malarial drugs. Define $Q^N(t)$ to be the probability that at least 1 of $N$ hypnozoites has relapsed by time $t$.

$$Q^N(t) = 1 - P_0^N(t) = 1 - \left( \frac{\mu + \alpha e^{-(\mu+\alpha)t}}{\mu + \alpha} \right)^N \quad (9)$$

Accounting for a geometrically distributed number of hypnozoites gives:

$$Q^{G(N)}(t) = \sum_{k=0}^{\infty} \frac{1}{N+1} \left( \frac{N}{N+1} \right)^k Q^k(t) \quad (10)$$

where $G(N)$ denotes a geometric distribution.

An individual $j$ followed up after a primary *P. vivax* infection will either relapse ($I_j = 1$) or avoid infection ($I_j = 0$). Denote $\tau_j$ to be the time of detection of infection, or if uninfected, the time until the end of follow up. The likelihood of the parameters $\theta = \{N, \alpha, \mu\}$ given the data $D_j = \{I_j, \tau_j\}$ is:

$$L(\theta | D_j) = \left( \frac{dQ^{G(N)}}{dt} \bigg|_{t=\tau_j} \right)^{I_j} \left( 1 - Q^{G(N)}(\tau_j) \right)^{1-I_j} \quad (11)$$

The log-likelihood ($LL$) for all $j$ individuals is:

$$LL = \sum_j \left( I_j \log \left( \frac{dQ^{G(N)}}{dt} \bigg|_{t=\tau_j} \right) + (1 - I_j) \log \left( 1 - Q^{G(N)}(\tau_j) \right) \right) \quad (12)$$

Data on time to first relapse were not sufficiently informative to estimate the three parameters simultaneously and hence prior distributions were assumed. $N$ was assumed to have an informative gamma prior

**Table 1.** Description of model parameters

| Parameter | Description | Value | Reference |
|---|---|---|---|
| Within-host | | | |
| $N$ | number of hypnozoites per infection | 8.5 | estimate* |
| $\alpha$ | rate of hypnozoite activation | 1/332 day$^{-1}$ | estimate* |
| $\mu$ | rate of hypnozoite/hepatocyte death | 1/425 day$^{-1}$ | estimate* |
| $d$ | duration of temperate long-latency | 180 days | (*Battle et al., 2014*) |
| $\sigma_d$ | standard deviation of temperate long-latency | 30 days | (*Battle et al., 2014*) |
| $M$ | number of compartments for simulating long-latency: $M = (d/\sigma_d)^2$ | 36 | |
| $\delta$ | rate of progression through long-latency compartments: $\delta = M/d$ | 0.2 day$^{-1}$ | |
| Humans | | | |
| $b$ | transmission probability: mosquito to human | 0.5 | (*Smith et al., 2010*) |
| $r$ | rate of clearance of blood-stage infections | 1/60 day$^{-1}$ | (*Collins et al., 2003*) |
| $f$ | relapse frequency (1/time to first relapse) | 1/76 day$^{-1}$ | *Equation 6* |
| $h$ | expected number of relapses | 4.7 | *Equation 4* |
| $\gamma$ | rate of hypnozoite clearance | 1/420 day$^{-1}$ | *Equation 5* |
| Mosquitoes | | | |
| $a$ | mosquito biting frequency | 0.21 day$^{-1}$ | (*Garrett-Jones, 1964*) |
| $g$ | mosquito death rate (1/mosquito life expectancy) | 0.1 day$^{-1}$ | (*Gething et al., 2011b*) |
| $m$ | number of mosquitoes per human | calculated | |
| $n$ | duration of sporogony in mosquito | 12 days | (*Gething et al., 2011b*) |
| $c$ | transmission probability: human to mosquito | 0.23 | (*Bharti et al., 2006*) |

*Based on estimates from South East Asian tropical strains.

distribution with median 10 (95% credible interval (CrI): 1, 28) (*Beier et al., 1991*). μ was assumed to have an informative gamma prior distribution with median 1/200 (95% CrI: 1/309, 1/140) day$^{-1}$ (*Malato et al., 2011*). α was assumed to have an uninformative uniform prior distribution U(0,1). The likelihood in equation (*Ishikawa et al., 2003*) was sampled using a Metropolis–Hastings Markov Chain Monte Carlo (MCMC) algorithm and the posterior parameter distributions estimated (*Figure 1—figure supplement 1*). The posterior median parameter estimates and 95% credible intervals are presented in *Supplementary file 1*.

## Acknowledgements

Bob Verity is thanked for contributions to the mathematical methods. This work was supported by a fellowship to MTW from the MRC. SIH is funded by a Senior Research Fellowship from the Wellcome Trust (#095066), which also supports KEB. ACG acknowledges support from the Bill and Melinda Gates Foundation and MRC Centre funding. IM is supported by an NHMRC Senior Research Fellowship (#1043345). SK was supported by an NHMRC Early Career Fellowship (#1052760). IM and SK acknowledge support from the Victorian State Government Operational Infrastructure Support and Australian Government NHMRC IRIISS.

## Additional information

### Competing interests

SIH: Reviewing editor, *eLife*. The other authors declare that no competing interests exist.

### Funding

| Funder | Grant reference number | Author |
|---|---|---|
| Medical Research Council | Population Health Scientist Fellowship | Michael T White |

| Funder | Grant reference number | Author |
| --- | --- | --- |
| National Health and Medical Research Council | Senior Research Fellowship | Ivo Mueller |
| National Health and Medical Research Council | Research Fellowship | Stephan Karl |
| Wellcome Trust | Senior Research Fellowship | Katherine E Battle, Simon I Hay |

The funders had no role in study design, data collection and interpretation, or the decision to submit the work for publication.

## Author contributions

MTW, Conception and design, Analysis and interpretation of data, Drafting or revising the article; SK, IM, ACG, Conception and design, Drafting or revising the article; KEB, SIH, Acquisition of data, Analysis and interpretation of data, Drafting or revising the article

## Author ORCIDs

Simon I Hay, 🔴 http://orcid.org/0000-0002-0611-7272

# Additional files

## Supplementary files

• Supplementary file 1. Estimated parameters for the within-host relapse.

• Source data 1. 'Individual_Level_Relapse_Data.xlsx': Data on time to first relapse infection from Battle et al Malaria Journal 2014, **13**:144.

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
