## [Decision Letter]

Thank you for sending your work entitled “Modelling the contribution of the hypnozoite reservoir to *Plasmodium vivax* transmission” for consideration at *eLife*. Your article has been favorably evaluated by Prabhat Jha (Senior editor), Mark Jit (Reviewing editor), and 3 reviewers, one of whom is a member of our Board of Reviewing Editors. Two of the reviewers have agreed to reveal their identity: Philip McQueen and Laith Yakob.

The Reviewing editor and the reviewers discussed their comments before we reached this decision, and the Reviewing editor has assembled the following comments to help you prepare a revised submission.

The reviewers and I felt that your report was interesting and addressed an important public health issue. The technical details were well-described despite being complex. The reviewers were mainly concerned with the plausibility of the biological assumptions behind the relapse model. Hence we expect you to either provide rigorous justification, or to modify, the following assumptions (perhaps by a closer examination of the existing data):

1) Relapses: The likelihood function ([Disp-formula equ10 equ11]) assumes that the relapse status of individual patients is known (I_j=0 or 1). However, the Results section discusses the importance of undetected relapses (relapses within 14 days of an earlier relapse) in producing an apparent periodicity in the temporal pattern of relapses, implying that I_j=0 during monitored time but is really equal to 1. If the effect of undetected relapses is expected to be a small correction for clinically relevant values of N and so can be ignored, then you need to state and justify that assumption in your formulation of [Disp-formula equ10 equ11].

2) Constant activation of hypnozoites in the liver: this does not appear to be consistent with evidence. If activation starts immediately, then it would not fit the 3 week intervals for tropical P. vivax after rapidly eliminated drugs. It also does not explain time clustering of relapse intervals or heterologous genotypes in relapses.

3) Constant death of hypnozoites: this implies random invasion that may well not be true.

4) In addition to these biological assumptions, the reviewers did not feel that Figure 2 showed any obvious periodicity in relapses. The argument would be strengthened by using more formal methods to establish this, e.g. plotting the distribution of relapses for large N (to visually establish multi-modality), and/or estimating the spectral density of the time series.

---

## [Author Response]

The responses to the comments below and the ensuing changes to the manuscript should provide additional evidence for the plausibility of the biological assumptions.

*1) Relapses: The likelihood function (*[Disp-formula equ10 equ11]*) assumes that the relapse status of individual patients is known (I_j=0 or 1). However, the Results section discusses the importance of undetected relapses (relapses within 14 days of an earlier relapse) in producing an apparent periodicity in the temporal pattern of relapses, implying that I_j=0 during monitored time but is really equal to 1. If the effect of undetected relapses is expected to be a small correction for clinically relevant values of N and so can be ignored, then you need to state and justify that assumption in your formulation of*
[Disp-formula equ10 equ11].

The reviewers make a germane point about the difference between detected and undetected relapses and highlights an area that could have been described more clearly in the original manuscript. We assume that a relapse can go undetected if it is suppressed by prophylaxis due to treatment of a previous infection or if it cannot be detected due to existing parasites. In Figure 2 it was assumed that if a hypnozoite was predicted to activate within 14 days of a previous infection, then the subsequent relapse would go undetected.

This point has prompted an additional clarification in the manuscript, namely that not every activating hypnozoite will correspond to a relapse. For example if two hypnozoites activate within a few days of each other then only one relapse would be counted. An adjustment to the axis labels in Figure 3 has been made to reflect this, with the axis now reading number “number of relapsing hypnozoites” instead of “number of hypnozoites”.

Note that the likelihood in [Disp-formula equ10 equ11] are for the first relapse only and so avoid the problem of a second relapse going undetected due to a previous relapse.

The text has been modified to clarify these points.

*2) Constant activation of hypnozoites in the liver: this does not appear to be consistent with evidence. If activation starts immediately, then it would not fit the 3 week intervals for tropical P. vivax after rapidly eliminated drugs. It also does not explain time clustering of relapse intervals or heterologous genotypes in relapses*.

To our knowledge there are no published studies of follow-up for recurrent *P. vivax* infections after rapidly eliminated anti-malarial drugs. Douglas *et al* review studies of the follow-up for recurrent *P. vivax* infections after the administration of ACTs, but in all cases the rapidly eliminated artemisinin component was administered alongside a partner drug with a longer half-life.

Chloroquine, which is slowly eliminated, is still widely used to treat *P. vivax*. In Battle *et al* several of the studies reviewed reported recurrent infections at day 14 despite the use of chloroquine. In areas with documented chloroquine resistance, Price *et al* report that treatment failure (defined via detectable parasites) occurs at a median of 14 days (range 3 to 28) days following chloroquine. However, the early detection of parasites could be due to either rapidly relapsing hypnozoites or the failure of chloroquine to eliminate all circulating blood-stage parasites.

Finally, in a recent study of the slowly eliminated drug dihydroartemisinin-piperaquine (DP), Tarning *et al* find recurrent *P. vivax* infections at day 14. A model where *P. vivax* recurrent infections occur in 3 week pulses is fitted to the data but it arguably provides no better fit than a model of constant activation of hypnozoites.

To properly test the hypothesis that relapses can occur in the first 2 to 3 weeks after primary infection would require treatment of *P. vivax* infections with artemisinin without a long-lasting partner drug and follow-up for recurrent low density infections. A new paragraph has been added to the discussion to reflect the points made here in response to the reviewers’ comment.

[15] Artemisinin combination therapy for *vivax* malaria. *Lancet Inf Dis.*

[4] Geographical variation in *Plasmodium vivax* relapse. *Malaria Journal.*

[42] Global extent of chloroquine-resistant *Plasmodium vivax*: a systematic review and meta-analysis. *Lancet Inf Dis.*

[53] Population pharmacokinetics and antimalarial pharmacodynamics of piperaquine in patients with *Plasmodium vivax* malaria in Thailand. *CPT Pharmacometrics Syst Pharmacol.*

*3) Constant death of hypnozoites: this implies random invasion that may well not be true*.

Once a hypnozoite is established in a liver hepatocyte, it is assumed to not invade other hepatocytes. The constant death rate of hypnozoites depends on the assumption of constant death of invaded liver hepatocytes (Malato *et al*).

[36] Fate tracing of mature hepatocytes in mouse liver homeostasis and regeneration. *J Clin Inv.*

*4) In addition to these biological assumptions, the reviewers did not feel that*
Figure 2
*showed any obvious periodicity in relapses. The argument would be strengthened by using more formal methods to establish this, e.g. plotting the distribution of relapses for large N (to visually establish multi-modality), and/or estimating the spectral density of the time series*.

This point is well taken. The plot below shows simulated patterns of relapses (tropical phenotype) for large N, ranging from 30 up to 200. For N = 200 there is a near constant activation of hypnozoites (red and grey triangles). However after censoring for undetected relapses (due to prophylaxis or existing parasitemia) the detected relapses (red triangles) appear periodic with period ∼ 14 days. This pattern is also evident for N = 30, 50, 100 (Figure 9).Author response image 1.

Figure 2 of the manuscript has been revised (now N=5, 10, 50 over a 12 month period) to more clearly show the periodic pattern.

A spectral analysis with simulated data would show increasing evidence of periodicity with increasing N. For very large N the period will approach 14 days. The period between consecutive relapses will be longer for smaller N. We believe that a full presentation of this analysis would go beyond the scope of what can be easily communicated to a wide audience.